Persona2vec: a flexible multi-role representations learning framework for graphs

Yoon Jisung 1 2
http://orcid.org/0000-0003-4627-9273 Yang Kai-Cheng 2
http://orcid.org/0000-0001-9590-3859 Jung Woo-Sung 1 3 4
http://orcid.org/0000-0002-4352-4301 Ahn Yong-Yeol 2 5 6 yyahn@iu.edu
1 Department of Industrial and Management Engineering, Pohang University of Science and Technology , Pohang , Republic of Korea
2 Center for Complex Networks and Systems Research, Luddy School of Informatics, Computing, and Engineering, Indiana University , Bloomington, IN , USA
3 Department of Physics, Pohang University of Science and Technology , Pohang , Republic of Korea
4 Asia Pacific Center for Theoretical Physics , Pohang , Republic of Korea
5 Connection Science, Massachusetts Institute of Technology , Cambridge, MA , USA
6 Network Science Institute, Indiana University , Bloomington, IN , USA
Shang Yilun
Electronic publication date: 2021 Mar 30
Publication date: 2021
Volume: 7
Electronic Location ID: e439
Received 2020 Dec 24; Accepted 2021 Feb 22
Copyright: © 2021 Yoon et al.
Copyright year: 2021
Copyright holder: Yoon et al.
License: This is an open access article distributed under the terms of the Creative Commons Attribution License, which permits unrestricted use, distribution, reproduction and adaptation in any medium and for any purpose provided that it is properly attributed. For attribution, the original author(s), title, publication source (PeerJ Computer Science) and either DOI or URL of the article must be cited.
License URL: https://creativecommons.org/licenses/by/4.0/

Keywords: Graph embedding, Overlapping community, Social context, Social network analysis, Link prediction

Funding: Air Force Office of Scientific Research FA9550-191-0391 This work is supported by the Air Force Office of Scientific Research under award number FA9550-191-0391. The funders had no role in study design, data collection and analysis, decision to publish, or preparation of the manuscript.

==============================
Graph embedding techniques, which learn low-dimensional representations of a graph, are achieving state-of-the-art performance in many graph mining tasks. Most existing embedding algorithms assign a single vector to each node, implicitly assuming that a single representation is enough to capture all characteristics of the node. However, across many domains, it is common to observe pervasively overlapping community structure, where most nodes belong to multiple communities, playing different roles depending on the contexts. Here, we propose persona2vec, a graph embedding framework that efficiently learns multiple representations of nodes based on their structural contexts. Using link prediction-based evaluation, we show that our framework is significantly faster than the existing state-of-the-art model while achieving better performance.

Introduction

Graph embedding maps the nodes in a graph to continuous and dense vectors that capture relations among the nodes (Perozzi, Al-Rfou & Skiena, 2014; Grover & Leskovec, 2016; Tang et al., 2015). Resulting node representations allow direct applications of algebraic operations and common algorithms, facilitating graph mining tasks such as node classification (Sen et al., 2008; Perozzi, Al-Rfou & Skiena, 2014), community detection (Fortunato, 2010; Yang et al., 2016), link prediction (Grover & Leskovec, 2016), visualization (Tang et al., 2015), and computer vision (Xie et al., 2020). Most methods map each node to a single vector, implicitly assuming that a single representation is sufficient to capture the full characteristics of a node.

However, nodes often play multiple roles. For instance, people have multiple roles, or “personas”, across contexts (e.g., professor, employee, and so on) (Ahn, Bagrow & Lehmann, 2010; Coscia et al., 2014; Leskovec et al., 2009; Leskovec, Lang & Mahoney, 2010). Similarly, proteins and other biological elements play multiple functionalities (Palla et al., 2005; Gavin et al., 2006; Ahn, Bagrow & Lehmann, 2010). Another example is the polysemy of words when their relations are modeled with graphs; many words possess multiple meanings differentiated by the contexts (Chen, Liu & Sun, 2014; Li & Jurafsky, 2015; Iacobacci, Pilehvar & Navigli, 2015). Explicit modeling of such multiplicity and overlapping clusters has been fruitful not only for community detection (Rosvall et al., 2014; Coscia et al., 2014; Epasto, Lattanzi & Paes Leme, 2017), but also for improving the quality of embedding (Li & Jurafsky, 2015; Epasto & Perozzi, 2019; Liu et al., 2019). Yet, with the scarcity of embedding methods embracing this idea, the full potential of this approach has not been properly explored.

In this paper, we propose persona2vec, a scalable framework that builds on the idea of ego-splitting (Epasto, Lattanzi & Paes Leme, 2017), the process of identifying local structural contexts of a node via performing local community detection on the node’s ego-network. For each detected local community (role), we transform each node into multiple personas if there are multiple local communities to which the node belongs. After the split, the original node is replaced by the new persona nodes that inherit the connection from each local community, producing a new persona graph. Instead of separating a node’s persona nodes from each other completely (Epasto & Perozzi, 2019), we add directed, weighted edges between personas to capture their origin. In doing so, we allow the direct application of the existing graph embedding methods. In addition, we take an approach of considering persona-based learning as fine-tuning of the base graph embedding, achieving both efficiency and balance between information from the original graph and the persona graph. Compared with the previous approach (Epasto & Perozzi, 2019), our framework is conceptually simpler to understand and practically easier to implement. Furthermore, it achieves better performance in the link prediction tasks while being much faster. We also would like to clarify that the primary purpose of persona splitting is not about obtaining multiple representations, each of which may be suited for a specific task; it is about teasing out multiple contexts that a single node may possess. In other words, even with a single task, we argue that learning multiple representations for some nodes is highly beneficial.

In sum, we would like to highlight that our approach (1) drastically lowers the barrier for combining existing algorithms with persona splitting, (2) significantly improves the efficiency of the ego-splitting approach, while (3) consistently excelling the previous state-of-the-art model in the link prediction task. Our implementation of persona2vec is publicly available at https://github.com/jisungyoon/persona2vec.

Related work

In addition to graph embedding, our work is closely related to the research of identifying overlapping communities in graphs. Various non-embedding methods such as link clustering (Ahn, Bagrow & Lehmann, 2010; Evans & Lambiotte, 2009), clique percolation (Palla et al., 2005), and mixed membership stochastic blockmodel (Airoldi et al., 2008) have been proposed. Another thread of works focuses on using local graph structure to extract community information (Coscia et al., 2014; Epasto et al., 2015; Epasto, Lattanzi & Paes Leme, 2017). Specifically, Epasto, Lattanzi & Paes Leme (2017) introduce the persona graph method for detecting overlapping communities in graphs, leveraging ego-network partition. The combination of ego-network analysis and graph embedding methods is still rare. An example is SPLITTER (Epasto & Perozzi, 2019), which we use as the baseline in this paper. Instead of constraining the relations between personas with a regularization term, we propose a simpler and more efficient way of adding persona edges to the graph.

Our work is also related to the word disambiguation problem in a word embedding. Recently, word embedding techniques (Mikolov et al., 2013a, 2013b; Pennington, Socher & Manning, 2014) have been extensively applied to various NLP tasks as the vectorized word representations can effectively capture syntactic and semantic information. Although some words have multiple senses depending on the context, the original word embedding methods only assign one vector to each word. Li & Jurafsky (2015) shows that embedding that is aware of multiple word senses and provides a vector for each specific sense does improve the performance for some NLP tasks. For this issue, some utilize the local context information and clustering for identifying word sense (Reisinger & Mooney, 2010; Wu & Giles, 2015; Neelakantan et al., 2015), some resort to external lexical database for disambiguation (Rothe & Schütze, 2015; Iacobacci, Pilehvar & Navigli, 2015; Camacho-Collados, Pilehvar & Navigli, 2016; Chen, Liu & Sun, 2014; Jauhar, Dyer & Hovy, 2015; Pelevina et al., 2017), while some combine topic modeling methods with embedding (Liu, Qiu & Huang, 2015; Liu et al., 2015; Cheng et al., 2015; Zhang & Zhong, 2016). We adopt the idea of assigning multiple vectors to each node in the graph to represent different roles as well as exploiting local graph structure for the purpose.

Proposed method: persona2vec

persona2vec creates a persona graph, where some nodes are split into multiple personas. We then apply a graph embedding algorithm to the persona graph to learn the embeddings of the personas (see Fig. 1). Let us explain the method formally. Let G = (V, E) be a graph with a set of nodes V and a set of edges E. |V| and |E| denote the number of nodes and edges respectively. Let f:v→Rd be the embedding function that maps a node v to a d-dimensional vector space (d << |V|).

Figure 1 Illustration of persona2vec framework.

(A) A graph with an overlapping community structure. (B) Graph embedding of the original graph is obtained first to initialize the persona embeddings. (C) Transform the original graph into a persona graph. Every edge in the original graph is preserved in the persona graph, while new directed persona edges with weight λkio are added between the persona nodes. (D) Graph embedding is applied to the persona graph. (E) The final persona embedding where each persona node has its own vector representation.

Refined ego-splitting

We adopt and refine the ego-splitting method (Epasto, Lattanzi & Paes Leme, 2017; Epasto & Perozzi, 2019). For each node in the original graph, we first extract its ego-graph, remove the ego, and identify the local clusters. Every cluster in the ego-graph leads to a new persona node in the persona graph (see Figs. 1A and 1C). For example, if we consider each connected component as a local community with a connected component algorithm, node C in the original graph belongs to two non-overlapping clusters {A,B} and {D,E,F} in its ego-graph. Given these two clusters, in the persona graph, C is split into C1 and C2 to represent the two roles in respective clusters. C1 and C2 inherit the connections of C from both clusters separately (see Fig. 1C). On the other hand, node A only belongs to one ego cluster {B,C}, so it does not split into multiple personas.

Any graph clustering algorithm can be employed for splitting a node into personas. The simplest algorithm is considering each connected component in the ego-network (sans the ego) as a cluster. This approach is fast and works well on sparse graphs. However, in dense graphs, ego-networks are more likely to form fewer connected components, thus other algorithms such as the Louvain method (Blondel et al., 2008), Infomap (Rosvall & Bergstrom, 2008), and label propagation (Raghavan, Albert & Kumara, 2007) would be more appropriate.

In previous studies, the personas get disconnected without retaining the information about their origin, creating isolated components in the splitting process (Epasto, Lattanzi & Paes Leme, 2017; Epasto & Perozzi, 2019). Because of this disconnectedness, common embedding methods could not be directly applied to the splitted graph. A previous study attempted to address this issue by imposing a regularization term in the cost function to penalize separation of persona nodes originating from the same node (Epasto & Perozzi, 2019).

Here, instead of adopting the regularization strategy, we add weighted persona edges between the personas, maintaining the connectedness between them after the splitting (see Fig. 1C). Because the persona graph stays connected, classical graph algorithms and graph embedding methods can now be readily applied without any modification. As we will show later, our strategy achieves both better scalability and better performance.

In the persona graph, we set the weights of the unweighted original edges as 1 and tune the strength of the connections among personas with λ. Persona edges are directed and weighted, with weight λkoi, where koi is the out-degree of the persona node after splitting (see Fig. 1C). Assigning weight proportional to koi helps the random walker exploring both the local neighbors and other parts of the graph connected to the other personas regardless of the out-degree koi.

Imagine node u, which is split into np personas. Consider one of the personas i with out-degree koi and persona edges with weight wi. Then the probability pi that an unbiased random walker at i visits neighbors connected with the original edges at the next step is

(1) kiokio+npwi.

If we set constant weight wi = λ, then

(2) pi=kiokio+npλ=11+npkioλ,

which depends on koi. A random-walker would not explore its local neighborhood if np >> koi, while the opposite happens when np << koi. Instead, assigning the weight proportional to koi, namely wi = λkoi, removes such bias because

(3) pi=kiokio+npλkio=11+npλ,

which is independent of koi. Our experiments also show that using the out-degree yields better performance than assigning the identical weight to each persona edge. Our algorithm for refined ego-splitting is described in Algorithm 1. Note that it can be generalized to the directed graphs.

Algorithm 1 Refined ego-splitting for generating the persona graph.

Case of the undirected graph.

Input: Original graph G(V, E); weight parameter λ; non-overlapping local clustering algorithm C	
Output: Persona graph GP(VP, EP); node to personas mapping V2P; persona to local cluster mapping	
P2C	
1: function REFEGOSPLIT(G(V, E), λ, C)	
2:   for each vo ∈ V do	
3:    Pvo←C(vo) ⊳ find local clusters of vo	
4:    for each p ∈ Pvo do	
5:     Create vp, and add to GP, V2P(vo) ⊳ create persona nodes for local clusters	
6:     P2C(vp) ← p	
7:   for each edge (vi, vj) in E do	
8:    w ← weight of edge	
9:    for each persona node vp in V2P(vi) do	
10:     for each persona node v′p in V2P(vi) do	
11:      if vi ∈ P2C(v′p) and vj ∈ P2C(vp) then	
12:       Add original edges (vp, v′p, w), (v′p, vp, w) to EP	
13:   ko ← out-degree sequence after adding original edges	
14:   for each vo ∈ V do	
15:     for each pair (vi, vj) in V2P(vo) do	
16:     Add persona edges (vi, vj, kio × λ), (vj, vi, kjo × λ) to EP	
17:   return GP(VP, EP), V2P, P2C	

Persona graph embedding

As explained above, any graph embedding algorithm that recognizes edge direction and weight can be readily applied to the persona graph. Although we use Node2vec as the embedding method here, other embedding methods can also be employed. We initialize the persona vectors with the vectors from the original graph before ego-splitting (see Fig. 1B) to leverage the information from the original graph structure. Persona nodes that belong to the same node in the original graph are thus initialized with the same vector. We then execute the embedding algorithm for a small number of epochs to fine-tune the embedding vectors with the information from the persona graph (see Fig. 1). Experiments show that usually only one epoch of training is enough.

We find that training the embedding on the persona graphs from scratch fails to yield comparable results. Instead, initializing the embedding with the original graphs, i.e., our present method, consistently improves the performance, suggesting that mixing the structural information from both the original graph and the persona graph is crucial. Our full algorithm is described in Algorithm 2.

Algorithm 2 persona2vec.

Our method for generating persona node embeddings.

Input:	
 G(V,E), Original graph	
 d, embedding dimension	
 γb, number of walks per node for base embedding	
 tb, random walk length for base embedding	
 wb, window size for base embedding	
 γp, number of walks per node for persona embedding	
 tp, random walk length for persona embedding	
 wp, window size for persona embedding	
 α, learning rate	
 REFEGOSPLIT, refined ego-splitting method	
 V2P, node to personas mapping	
 EMBEDDINGFUNC, a graph embedding method e.g. DeepWalk, Node2vec	
Output:	
 ΦGP, a NP × d matrix with d-dimensional vector representations for all NP persona nodes	
1: function PERSONA2VEC(G, d, γb, tb, wb, gp, tp, wp, REFEGOSPLIT, EMBEDDINGFUNC, α)	
2:   GP, V2P ← REFEGOSPLIT(G)	
3:   ΦG ← EMBEDDINGFUNC(G, d, γb, tb, wb, α)	
4:   for each vo ∈ V do	
5:    for each persona node vp in V2P(vo) do	
6:     ΦGP (vp) = ΦG(vo)	
7:   ΦGP ← EMBEDDINGFUNC(Gp, γp, tp, wp, α, ΦGP)	
8:   return ΦGP	

Complexity

Space complexity

The persona graph is usually larger than the original graph, but not too large. Node u with degree ku may be split into at most ku personas. In the worst case, the number of nodes in the persona graph can reach O(|E|). But, in practice, only a subset of nodes split into personas, and the number of personas rarely reaches the upper bound. If we look at the persona edges, for a node u with degree ku, at most O(ku2) new persona edges may be added. Thus, the whole persona graph has at most O(|V|×kmax2) or O(|V|3) (∵ kmax ≤ |V|) extra persona edges. If graph’s degree distribution follows a power-law distribution P(k) ∼ k−γ, then kmax ∼ |V|1/γ−1. Hence, it could be O(|V|γ+1/γ−1) and it is between O(|V|2) and O(|V|3) (∼ 2 ≤ γ ≤ 3 in general). However, real graph tends to be sparse and ki << |V|. If we further assume ki<|E| holds for every node, then ∑n=1|V|kn2≤∑n=1|V|kn|E|=2|E||E|. Under this assumption, the upper bound becomes O(|E|3/2). Similarly, with the scale-free condition, the upper bound could be O(|E||V|1/γ−1), which is between O(|E||V|1/2) and O(|E||V|). Again, in practice, the number of persona edges is much smaller than this upper bound. To illustrate, we list the number of nodes and persona edges in the persona graph for the graphs we use in this paper in Table 1. All considered, the extra nodes and edges do not bring too much space complexity burden in practice.

Table 1 Descriptive statistics of the graphs used in the evaluation.

We report the number of nodes |V|, number of edges |E|, number of nodes in the persona graph |Vp|, the ratio of |Vp| over |V|, number of persona edges |Ep| added in ego-splitting, and the ratio of |Ep| over |E3/2| which is the upper bound of space complexity.

Dataset	Type	|V|	|E|	|Vp|	|Vp|/|V|	|Ep|	|Ep/E2/3|	
PPI	Undirected	3,863	38,705	16,734	4.34	132,932	0.0175	
ca-HepTh	Undirected	9,877	25,998	16,071	1.86	33,524	0.0800	
ca-AstroPh	Undirected	17,903	197,301	25,706	1.44	29,102	0.0003	
Wiki-vote	Directed	7,066	103,633	21,467	3.04	118,020	0.0035	
Soc-epinions	Directed	75,877	508,836	220,332	2.90	3,550,594	0.0098	

Time complexity

Assessing the time complexity requires consideration of the two steps: ego-splitting and embedding. The ego-splitting algorithm has complexity of O(|E|3/2+|E|T(|E|)) in the worst case, where |E| is the number of edges in the original graph and T(|E|) is the complexity of detecting the ego clusters in the graph with |E| edges (Epasto, Lattanzi & Paes Leme, 2017). The embedding on the persona graph, which dominates the whole embedding procedure, has complexity O(|Vp|γ twd(1 + log(|Vp|))) which is time complexity of Node2vec, where |Vp| is the number of nodes, γ is the number of random walkers, d is the embedding dimension, and w is the window size (Chen et al., 2018).

The final complexity is O(|E|3/2+|E|T(|E|))+O(|V|γtwd(1+log⁡(|V|))). Removing the constant factors and assuming close-to-linear local community detection algorithm, the whole process has time complexity of O(|E|3/2) with space complexity of O(|E|3/2) if ki<|E| holds. Complexity can be increased depending on the clustering algorithms on the ego-network.

To test the validity of our assumptions, we sample 1,000 graphs from a public network repository (Rossi & Ahmed, 2015). We apply the refined ego-splitting with connected component algorithms on these samples and report the actual number of persona edges |Ep| with respect to the practical upper bound |E|3/2 in Fig. 2, which shows that the actual number of persona edges |Ep| rarely exceeds the tighter upper bound that we propose and is usually orders of the magnitude smaller.

Figure 2 Comparison of the the number of persona edges |Ep| to the practical upper bound |E|3/2.

Optimization

Any kind of graph embedding method can be considered, for simplicity, we choose the classical random-walker based embedding method (e.g., Node2Vec, DeepWalk). In the model (Perozzi, Al-Rfou & Skiena, 2014), the probability of a node vi co-occurring with a node vj is estimated by

(4) p(vi|vj)=exp⁡(Φ′vi⋅Φvj)∑k=1Vexp⁡(Φ′vk⋅Φvj),

where Φvi and Φ′vi are the ‘input’ and ‘output’ embedding of node i. We use input embedding Φ which is known to be more useful and more widely used. Denominator of Eq. (4) is computationally expensive (Yang et al., 2016; Cao, Lu & Xu, 2016) and there are two common approximations: hierarchical softmax (Morin & Bengio, 2005) and negative sampling (Mikolov et al., 2013b). We adopt negative sampling not only because it is simpler and popular but also because it shows better performance.

Case study

Before diving into systematic evaluations, we provide two illustrative examples: Zachary’s Karate club network and a word association network.

Case study: Zachary’s Karate club network

We use Zachary’s Karate club network (Zachary, 1977), a well-known example for the community detection. Nodes represent members of the Karate club, and edges represent ties among the members (see Fig. 3A). Although it is often considered to have two large disjoint communities, smaller overlapping communities can also be seen, highlighted by nodes such as 1, 3, 28, and 32. In Fig. 3B, we present the persona graph of the network. persona2vec successfully recognizes these bridge nodes and places their personas in reasonable locations. Take node 1 for example. It splits into four persona nodes, which then end up in two different communities. The orange and green communities are clearly separated as a result. We also show the ten predictions with the highest scores from the link prediction experiment in Fig. 3D and ensure that the model predicts missing edges well.

Figure 3 Case Study: Zachary’s Karate club network.

(A) The Zachary’s Karate club network with the force-atlas layout (Zachary, 1977). Nodes are colored by communities detected by the Louvain modularity method (Blondel et al., 2008). (B) The persona graph. Nodes are colored by k-means clusters (MacQueen, 1967) from the embedding vectors. Coordinates of the persona nodes come from the 2-D projection of the embedding with t-SNE (Maaten & Hinton, 2008). Light gray lines represent the persona edges. (C) The network with 20% of edges (16 edges) removed for the link prediction experiment. (D) The network with ten predictions with the highest scores from the link prediction experiment. Blue links represent correctly predicted edges and red edges indicate incorrectly predicted ones.

Case study: word association network

Word association network captures how people associate words together (free association task). The dataset was originally assembled from nearly 750,000 responses from over 6,000 peoples. Participants were shown 5,019 words and asked to write down the first word that sprang in mind and all the word pairs were collected with their frequency as the weights. This dataset forms a weighted, directed graph of words that captures their multiple senses. Although it is, in principle, possible to run our method on the original graph, for simplicity, we convert it into an undirected, unweighted graph by neglecting weight and direction (Ahn, Bagrow & Lehmann, 2010). In Fig. 4, we show the persona2vec clusters around the word “Newton”. We use the Louvain method (Blondel et al., 2008) to split the personas of each word. persona2vec successfully captures multiple contexts of the word “Newton”. For instance, the red persona is associated with “scientists” and “philosopher”, the gray one is linked to the physics, and the yellow one is associated with “apple” (note that there is a cookie called “(Fig) Newton” in the U.S.). Furthermore, persona2vec also captures different nuances of the word “law” that are related to the crime (brown cluster) and the legal concepts (orange cluster).

Figure 4 The word association network, clusters around the word “Newton”.

Coordinates of the words come from the 2-D projection of the embedding vectors with UMAP (McInnes, Healy & Melville, 2018). Word colors correspond to the clusters obtained by k-means clustering (MacQueen, 1967) on the embedding vectors.

Numerical experiment

Link prediction task

To systematically evaluate the performance and scalability of the persona2vec framework, we perform a link prediction task using real-world graphs (Grover & Leskovec, 2016; Abu-El-Haija, Perozzi & Al-Rfou, 2017). Link prediction aims to predict missing edges in a graph with partial information, which is useful for many tasks such as suggesting new friends on social networks or recommending products. It has been employed as a primary task to evaluate the performance of unsupervised graph embedding methods (Abu-El-Haija, Perozzi & Al-Rfou, 2017; Zhang et al., 2018).

We follow the task setup from the literature (Grover & Leskovec, 2016; Abu-El-Haija, Perozzi & Al-Rfou, 2017). First, the edge set of an input graph is divided equally and randomly into Etrain and Etest. We then refine Etest using a rejection sampling method based on the criterion that, even when we remove all edges in Etest, the graph should be connected as a single component. Etrain is used to train the models, and edges in Etest are used as positive examples for the prediction task. Second, a negative edge set E(−) of non-existent random edges with the same size of Etest are generated to provide negative examples for testing. The performance of a model is measured by its ability to correctly distinguish Etest and E(−) after being trained on Etrain. We then report ROC-AUC.

Datasets

To facilitate the comparison with the state-of-the-art baseline, we use five graph datasets that are publicly available and previously used (Epasto & Perozzi, 2019; Leskovec & Krevl, 2014). We summarize them as follows.

PPI is a protein-protein interaction graph of Homo sapiens (Stark et al., 2006). Nodes represent proteins and edges represent physical interactions between the proteins. ca-HepTh is a scientific collaboration graph. It represents the co-authorship among researchers from the Theoretical High Energy Physics field, derived from papers on arXiv. ca-AstropPh is also scientific collaboration graph, but from Astrophysics. wiki-vote is a voting network, each node is a Wikipedia user and a directed edge from node i to node j represents that user i voted for user j to become an administrator. soc-epinions is a voting graph from a general consumer review site Epinions.com, each node is a member, and a directed edge from node i to node j means that member i trusted member j.

We use the largest connected component of the undirected graphs and the largest weakly connected component of the directed ones. The statistics of all the graphs are reported in Table 1.

Methods

The state-of-the-art method in this link prediction task is SPLITTER (Epasto & Perozzi, 2019), which also models multiple roles. As reported in the paper, it outperforms various exiting algorithms ranging across non-embedding methods like Jaccard Coefficient, Common Neighbors, and Adamic-Adar as well as embedding methods like Laplacian EigenMaps (Belkin & Niyogi, 2002), Node2vec (Grover & Leskovec, 2016), DNGR (Cao, Lu & Xu, 2016), Asymmetric (Abu-El-Haija, Perozzi & Al-Rfou, 2017) and M-NMF (Wang et al., 2017).

Given the state-of-the-art performance of SPLITTER, for simplicity, we compare our framework with SPLITTER using the identical task setup and datasets. In addition, because our method can be considered as an augmentation of a single-role embedding method, and because we use Node2vec as the base embedding method, we also employ Node2vec. We run the link prediction task using the original authors’ implementation of Node2vec and SPLITTER. The parameters are also kept consistent with the original paper.

persona2vec and SPLITTER have multiple representations on each node, which leads to non-unique similarity estimations between two nodes. Hence, we define the similarity score of a pair of nodes on persona2vec as the maximum dot-product of embedding vectors between any pair of their personas. We found that, among experiments with three aggregation functions min, max, mean, the highest performance is achieved with max, the same with SPLITTER (Epasto & Perozzi, 2019). For SPLITTER, we use maximum cosine similarity, following the author’s note in their implementation.

Node2vec (baseline method)

For Node2vec, we set random walk length t = 40, the number of walks per node γ = 10, random walk parameters p = q = 1, the window size w = 5, and the initial learning rate α = 0.025. In the original paper, they learn an additional logistic regression classifier over the Hadamard product of the embedding of two nodes for the link prediction. In general, the logistic regression classifier improves the performance. Here, we report results on Node2vec with both dot products and the logistic regression classifier.

SPLITTER (baseline method)

For SPLITTER, we use the same parameters in the paper (Epasto & Perozzi, 2019) and aforementioned Node2vec baseline. We use Node2vec with random walk parameters p = q = 1.

persona2vec (our proposed method)

We set the hyper-parameters of the original graph embedding with tb = 40, γb = 10, wb = 5. For the persona embedding, we set tp = 80, γp = 5, wp = 2 to better capture the micro-structure of the persona graph. The size of the total trajectories is determined by the random walk length t* times the number of walks per node γ*, so we keep t*γ* constant to roughly preserve the amount of information used in the embedding. For both embedding stages, we use α = 0.025, and Node2vec with the random walk parameters (p = q = 1) as the graph embedding function.

Experiment results

Figure 5 shows the link prediction performance of persona2vec in comparison with the baselines. Overall, persona2vec yields superior performance across graphs and across a range of hyperparameter choices. We show that augmenting Node2vec by considering personas significantly improves the link prediction performance, evinced by the significant performance gain (see Table 2).

Figure 5 Performance of persona2vec in the link prediction task.

We report the link prediction performance for each graphs for (A) PPI, (B) ca-HepTh, (C) ca-AstroPh, (D) wiki-vote, and (E) SOC-epinions. Number of epochs n is set to 1 in all experiments for persona2vec. Darker colors represent higher embedding dimensions. The confidence intervals are all within the range of the markers. Given the same number of dimensions, persona2vec is always on par with or better than SPLITTER.

Table 2 Performance of persona2vec with λ = 0.5.

All methods use d = 128. Node2vec* refers to Node2vec with the logistic regression classifier, SPLITTER* refers to SPLITTER with one epoch, and persona2vec* refers persona2vec with λ = 0.5, our suggested default. Performance gain is performance difference between persona2vec* and Node2vec. We omit the standard error which is smaller than 10−3. Bold numbers represent the best performance.

Method	PPI	ca-HepTH	ca_AstroPh	wiki-vote	soc-epinions	
Node2vec	0.585	0.825	0.901	0.694	0.547 ± 0.007	
Node2vec*	0.662 ± 0.001	0.848	0.914	0.705 ± 0.001	0.767 ± 0.002	
SPLITTER	0.856	0.903	0.982	0.931	0.961 ± 0.001	
SPLITTER*	0.853	0.898	0.984	0.931	0.954 ± 0.001	
persona2vec*	0.879	0.927	0.985	0.936	0.961	
Performace_gain	0.294	0.102	0.084	0.242	0.414 ± 0.007	

As expected, larger dimensions lead to better performance, although persona2vec achieves reasonable results even with tiny embedding dimensions like 8 or 16. We also show how the performance of persona2vec varies with λ. For undirected graphs, larger λ is beneficial but the trend saturates quickly. For directed graphs, however, optimal performance is achieved with smaller values of λ. In practice, we suggest starting with λ = 0.5 as a default parameter because the overall variation brought by λ is not substantial and even when the performance increases with λ, near-optimal performance can be achieved at λ = 0.5.

When compared with the SPLITTER baseline, persona2vec shows on par or better performances given the same embedding dimensions across a wide range of λ. We also report the performance summary for persona2vec with λ = 0.5 (our suggested default) compared with the best baselines in Table 2, which shows that persona2vec outperforms the baseline consistently. Also, we report the performance gain of persona2vec from Node2vec, because we used Node2vec as the base embedding method and persona2vec can be considered as an augmentation or fine-tuning of the base Node2vec vectors with local structural information. As shown, the persona-based fine-tuning significantly improves the performance.

We also study the effect of different optimization methods, i.e., hierarchical softmax and negative sampling in Fig. 6. We also find that cosine similarity consistently yields a better result with hierarchical softmax while dot product yields a better result with negative sampling regardless of the embedding methods. So, we use cosine similarity for hierarchical softmax and use dot product for negative sampling. Our experiments suggest that persona2vec tends to perform better with negative sampling while SPLITTER works better with hierarchical softmax. Nevertheless, persona2vec yields the best performance consistently.

Figure 6 Comparison of link prediction performance between persona2vec and SPLITTER with different approximations.

We report the link prediction performance across optimization methods for each graphs for (A) PPI, (B) ca-HepTh, (C) ca-AstroPh, (D) wiki-vote, and (E) SOC-epinions. HS refers to the hierarchical softmax and NS refers to the negative sampling. The star marker indicates the best link prediction performance.

In addition to the performance of the link prediction task, we also report the execution time of persona2vec and SPLITTER to compare their scalability in practice (see Fig. 7). Note that the reported execution time is from the link-prediction task, with half of the edges removed from the original graph. SPLITTER runs the embedding procedures for 10 epochs by default in the original implementation, whereas persona2vec only runs for one epoch. For a fair comparison, we also report the results of SPLITTER with one epoch of training. When being limited to only one epoch, SPLITTER’s performance slightly suffers on three graphs while it goes up or stays stable for the other two.

Figure 7 Comparison of elapsed time between persona2vec and SPLITTER.

Speed gains by persona2vec are shown.

Nevertheless, persona2vec is more efficient—39 to 58 times faster than SPLITTER with 10 epochs and five to eight times faster than SPLITTER with one epoch. The most likely reason behind the drastic difference is the overhead from the extra regularization term in the cost function of SPLITTER, which persona2vec does not need. In sum, persona2vec outperforms the previous state-of-the-art method both in terms of scalability and link prediction performance.

Conclusions

We present persona2vec, a framework for learning multiple node representations considering the node’s local structural contexts. persona2vec first performs ego-splitting, where nodes with multiple non-overlapping local communities in their ego-networks are replaced with corresponding persona nodes. The persona nodes inherit the edges from the original graph and remain connected by newly added persona edges, forming the persona graph. Initialized by the embedding of the original graph, the embedding algorithm applied to the persona graph yields the final representations. Instead of assigning only one vector to every node with multiple roles, persona2vec learns a vector for each of the personas. With extensive link prediction evaluations, we demonstrate that persona2vec achieves the state-of-the-art performance while being able to scale better. Moreover, our method is easy to comprehend and implement without losing any flexibility for incorporating other embedding algorithms, presenting great potential for applications. The possible combination with various algorithms provides vast space for further exploration. For instance, in a multi-layer network, inter-layer coupling connection can be interpreted as natural persona edges, and persona2vec may be applied to tackle the multi-layer link prediction problem.

The graph (relational) structure is ubiquitous across many complex systems, including physical, social, economic, biological, neural, and information systems, and thus fundamental graph algorithms have far-reaching impacts across many areas of sciences. Graph embedding, in particular, removes the barrier of translating methods to the special graph data structure, opening up a powerful way to transfer existing algorithms to the graphs and relational data. Furthermore, given that it is natural to assume that overlapping clusters and their heterogeneous functionality exist in most real networks, multi-role embedding methods may find numerous applications in physical, biological, and social sciences.

For their comments, we thank Sadamori Kojaku, Alessandro Flammini, Filippo Menczer, Xiaoran Yan, Filipi Nascimento Silva, and Minwoo Ahn.

Additional Information and Declarations

Competing Interests

Author Contributions

Data Availability

The authors declare that they have no competing interests.

Jisung Yoon conceived and designed the experiments, performed the experiments, analyzed the data, performed the computation work, prepared figures and/or tables, authored or reviewed drafts of the paper, and approved the final draft.

Kai-Cheng Yang conceived and designed the experiments, authored or reviewed drafts of the paper, and approved the final draft.

Woo-Sung Jung conceived and designed the experiments, authored or reviewed drafts of the paper, and approved the final draft.

Yong-Yeol Ahn conceived and designed the experiments, authored or reviewed drafts of the paper, and approved the final draft.

The following information was supplied regarding data availability:

The prepossessed version of PPI is available at Stanford University: https://snap.stanford.edu/node2vec/.

Other graphs (ca-AstroPh, ca-HepTh, wiki-Vote, soc-Epinions1) are also available at the SNAP library:

http://snap.stanford.edu/data/index.html.

Code is available at GitHub:

https://github.com/jisungyoon/persona2vec.

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
