# Peer review of "Persona2vec: a flexible multi-role representations learning framework for graphs"

_PeerJ Computer Science, doi:10.7717/peerj-cs.439_

## Round 0.1 · original submission · Minor Revisions

Please address the reviewers' comments and provide point to point responses. However, you should not defer to the reviewer if they indicated their own papers. Your revised version will not be sent to them for re-evaluation.

Reviewer 1 ·

Basic reporting

The paper is mostly well written, there are a couple of minor typos (line 101 "explores" should be "exploring", line 151 "shown" should be "shows"). Nothing that a simple proofread won't fix. The literature references are adequate. Figures and tables are of acceptable quality. Overall, the article of is of good quality.

One thing I'd improve is in presenting the idea of multiple node embeddings more clearly. Specifically, it should be clearer the difference between this method and simply building different embeddings with different methods. In both cases we generate different vectors for the same node, but in persona2vec the different vectors aid the *same* task, while in the alternative, you build different vectors specialized to solve *different* tasks (structural equivalence vs clustering, for instance).

Experimental design

The experiments show some standard evaluation techniques for link prediction. I would love to see a couple of examples of predicted links -- picking a network and show the ten predictions with highest score and whether the nodes are actually connected. This is rarely done in link prediction papers, though.

Data & code is publicly available, which is a plus.

Validity of the findings

The main contribution vs the state of the art is the addition of persona edges and their weighting scheme. While this is shown to be effective in practice, I wonder whether this is significant enough. The improvement is statistically significant as shown in Table 2, but also relatively small in absolute terms.

Additional comments

In general, I'm happy with the paper. I'm suggesting major revision instead of minor only for one reason, which I think might significantly impact some methods and/or require some additional experiments. Specifically: could the author use persona2vec for multilayer link prediction? In this case, interlayer coupling connections would provide some natural persona edges, but it'd be interesting to use a persona-like edge addition and weighting scheme to them (if we have n layers, do we connect all layers to all other layers in a clique fashion, or could we ignore some connections? Which weight should each coupling link have? Does it depend on the degree of the persona in a give layer?). There are some embedding based multilayer link predictors, along with some non-embedding ones.

Reviewer 2 ·

Basic reporting

1. The overall writing is good, however, there exist some typos, e.g.,

it show --> it shows in L158, model -> models in L28, please proofread the whole paper again to modify them accordingly.

2. The whole organizations of the paper are confusing, e.g. the related work is before the conclusion. I would advise the authors to move it after the introduction section.

3. Regarding the Persona graph embedding section, the authors should show more details, from the current version, I cannot see any detailed formulas w.r.t. this section.

4. A missed reference for graph embedding learning in CV community is 'region graph embedding for zero-shot learning published in ECCV20'

Experimental design

Experimental design is clear and sufficient.

Validity of the findings

The overal novelty is neat, e.g., the idea of using multiple features for one node is new.

One question is how to extend this idea to GCN formulation like in 'region graph embedding for zero-shot learning published in ECCV20'?

Additional comments

A good paper with minor langurage issues, and other minor drawbacks.

---

## Round 0.2 · accepted · Accept

The paper can be accepted.

Reviewer 1 ·

Basic reporting

All mostly good, see general comments.

Experimental design

All mostly good, see general comments.

Validity of the findings

All mostly good, see general comments.

Additional comments

Thank you for addressing my comments. For most of the changes, I have no further comments.

One minor thing: in the Zachary network, panel d, I initially counted only 9 blue+red links, instead of 10. I needed a 200% zoom to find the tenth (the short blue edge from node 1). Perhaps make it a little bit more obvious!

Re: the significance of the results: I actually never doubted the results were statistically significant! I was just wondering if the improvement is noticeable in absolute terms. In any case, this is super minor so I have no further things to say about it.

Reviewer 2 ·

Basic reporting

The same as last round.

Experimental design

The same as last round.

Validity of the findings

The same as last round.

Additional comments

The authors have addressed my concerns in this version. I would accept this paper accordingly.